# Glass—A Material Practice in the Anthropocene

**Inge Panneels** 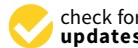

National Glass Centre, The University of Sunderland, Sunderland SR1 3SD, UK;
inge.panneels@sunderland.ac.uk

**Abstract:** This paper details and discusses *Material Journey* (2018), an art project by the author that was exhibited at the National Glass Centre in Sunderland (UK) from 9 June to 2 September 2018. This research project sought to interrogate the material impact of one art project made of glass by carefully considering the different stages of making—from design to production to the exhibition phase. The carbon footprint of an energy intensive material such as glass is often considered anathema to sustainable making practices in the field of applied arts. Whilst this paper makes the case that the material impact of individual art practices is negligible in the global context of carbon footprints, it nevertheless argues that the craft of 'making' has a critical role to play in the Anthropocene. Critically, this project is one of the first art projects in glass that critically examines the carbon footprint of a material practice. It is argued that this conversation is long overdue but makes the case that the tools for understanding and calculating the carbon footprint of a material practice are currently lacking and need more development.

**Keywords:** glass; making; crafts; applied arts; Anthropocene; sustainability; carbon footprint

---

## 1. Making in the Anthropocene

The concept of the Anthropocene (Crutzen and Stoermer 2000) has generated much debate in the Arts and Humanities in the last decade. The discussion on whether the earth has entered a new geological epoch is still to be decided by the International Commission of Stratigraphy but the term—the Anthropocene—has given a conceptual framework that has enabled a wide ranging and diverse discourse to unfold on the impact of human activity on the planet and its ecosystems (Chandler 2018; Lewis and Maslin 2018).

The mission of *Material Journeys* was to map the carbon footprint of one art project. *Craft in an Age of Change* (Crafts Council 2012), the Crafts Council report that examined the role of craft making in the second decade of the 21st century, noted some key trends in the sector. It noted that craft making skills and knowledge are valued in an increasingly skills-based economy. Secondly, it considered the impact of the digital on the sector where the hand-made is an assumed de-facto requirement. Thirdly, it proposed that the portfolio career model prevalent in the sector might provide a good role model for the 21st century where a career for life is no longer relevant for most. Lastly, and important to this article, the report noted that the craft sector has an environmental awareness and sensitivity. The craft sector, it noted, appears to be well placed to 'benefit from these wider shifts' (p. 10) in culture towards a more localised economic model based on more locally produced, ethically sourced and sustainable artisan products. However, the report also noted that some of the most significant craft disciplines, such as glass and ceramics, use production methods that are not environmentally friendly as they require large energy input for firing kilns and furnaces and use materials that have an environmental impact such as glazes and mould materials. Their environmental footprint is large—it is both energy intensive and in terms of Anthropocene thinking, has a geological timeframe that far outstrips other more sustainable materials. Glass produced by the Ancient Egyptians and Romans graces our museums

and galleries precisely because it lasts. Conversely, this is the argument that the glass packaging industry is currently making by promoting glass as a sustainable and 'green' packaging material, in comparison with plastics certainly, as a material that can be re-used and 'infinitely' recycled[1]. In her groundbreaking book *Doughnut Economics* (Raworth 2017), the economist Kate Raworth argued for seven new ways to think about economics that put both people and the planet at the heart of a radical new thinking about the economy. The doughnut model, rooted in feminist, gender, race and environmental theory, argued that the economic model itself needs to be re-designed as it is not fit for purpose for the 21st century. In relation to the environment, Raworth argued that economic orthodoxy has long portrayed a clean environment as a luxury good as explained in the Environmental Kuznets Curve that argued that pollution had to get worse before it can get better and that economic growth will (eventually) clean it up. But as Raworth noted, there is no such environmental law but simply degenerative industrial design. Instead, she argued for the need to create a circular economy that needs to take account of the whole loop—from raw material, to product, use and 'waste' into an integrated and purposefully designed, circular system. Whilst economists have started to place a value on the natural capital Nature provides (e.g., clean air, water, forests, fields, pollinating bees etc.), the calculation of a carbon footprint—the value given to the amount of carbon created in the production of goods and services—entered the public discourse. Neither system is without its critics, as both remain focused on monetary values. However, they have arguably opened up an important discourse around the material impact of resources and their consideration within a potentially new economic model as proposed by Raworth.

The philosopher Gilles Deleuze and psychoanalyst Félix Guattari (Deleuze and Guattari 2004) argued to 'follow the materials' as artisans or practitioners, where the 'dance' of materials leads the makers and vice versa. 'As the artisan thinks *from* materials, so the dancer thinks from the body' (Ingold 2013, p. 94). The anthropologist Tim Ingold (2013) observed that the physical properties of a material are irrevocably enmeshed in human understanding of materials as experienced, not only in cultural artefacts but in cultural understanding and appropriation of materials. In other words, the two-side nature of materiality defines our understanding of the material world. As such, he argued, the vitality of a work of art lies in its materials and as such is never 'finished'. The archeologist Bjørnar Olsen observed that 'Things are more persistent than thought. They evidently last longer than speech or gestures. Things are concrete and offer stability' (Olsen in Ingold 2013, p. 102). The material legacy of 'things' thus needs to be considered. As noted earlier, our material history resides in the collections of national museums—the Egyptian glass perfume bottle or the Roman glass vase, speak of the material culture of their time. This then begs the question, how will the materials produced in our time, reflect our material culture? The archaeologist Joshua Pollard observed how contemporary environmental artists have challenged our assumptions about the durability of 'things' by producing art works that are temporary or ephemeral in nature, quoting the practice of Andy Goldsworthy's temporal artworks (in Ingold 2013, p. 104). But of course, not all artwork can, or should, be made from temporary materials or made in fleeting moments. It is within this context that the *Material Journey* project thus needs to be considered. Therein, then, lies the dilemma of considering an art practice rooted in the materiality of glass making, of a material that speaks of long geological timescales and large carbon footprints. How, then, should this practice be framed in the Anthropocene?

## 2. *Material Journey*—The Material Journey of an Artwork

It has been established that glass has played a critical role in Modernity (Macfarlane and Martin 2002—Panneels 2019) as a key material that allowed *observation*, a key tenet of modern science in particular—observation of that which was not discernible to the naked eye but made visible through lenses in microscopes or telescopes. Glass became a material that spoke of Modernity, in industrial

---

[1]　There are caveats with this in that there is always a percentage of virgin material needed to make a new glass batch.

design and in architecture, and also played a critical role in the expansion of communication to name but a few of its key contributions to the Modern Era. How then, should glass be considered in the Anthropocene, which is considered—but still contested—as the end of modernity (Chandler 2018).

During nearly two decades of glass making as an art practice, my making skills as a craft maker have focused on kiln-based techniques and practices, rather than glass blowing or stained glass, which are perhaps better known traditional glass techniques. Kiln glass methods rely on sculpting techniques, of modelling forms from clay for example, then creating a plaster mould from this model into which the glass is cast. *Material Journey* looked at the distinct processes involved in the making of a cast glass object. The motif of a small glass cast paper boat had been used in previous works as a symbol of the human journey and exploration such as *Off The Map* (2015), *Arctic* (2015) and *Map-i: Buchan Way* (Figure 1). *Material Journey* was thus to consider the production of an artwork, mainly from the studio based in the Scottish Borders and its subsequent exhibition at the National Glass Centre in Sunderland. Conceptually, the work unconsciously borrowed from the work of Scottish artist George Wyllie whose *Paper Boat* (1989) lamented the decline of the ship building industry on the River Clyde (Patience and Wyllie 2016) and can thus be translated to the River Wear.

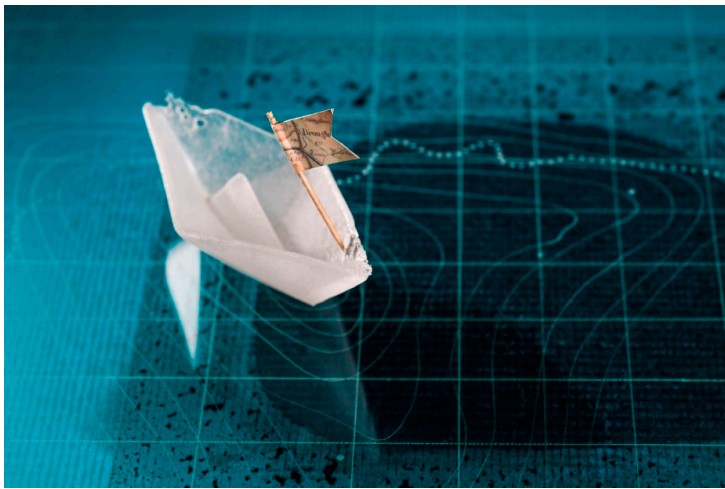

**Figure 1.** *Map-i: Buchan Way, 2015*—with a cast glass paper boat.

The small paper boat thus became emblematic of the material journey of the project—from idea, to work in progress to a final 'finished' artwork. The project broke down the practice of producing an artwork in three distinct phases—the Thinking Phase, the Making Phase and finally the Exhibiting Phase. The project considered the many different forms of energy embodied in the work—of thinking power (designing, drawing, problem solving and research), manual making power (shaping, forming, wiping, grinding, polishing, cleaning, etc.) as well as the embodied energy in the material itself. This journey 'took us down the rabbit hole . . . of carbon footprint calculation . . . as someone said . . . know the beginning before the end . . . ' The results were collated in a poem-like text, as text has often been used in previous works (see Supplementary 1). I subsequently discovered that Newton and Helen Harrison, pioneers of the environmental art movement, often use poetic text besides their visuals as an integral part of the work. This methodology was devised to 'answer the question of how much information you could compress and in how short a reading time for understanding to take place of extremely complex eco-political observations, leading ultimately to [ . . . ] understanding' (Harrison et al. 2016, p. 67).

*Material Journey* (Figure 2: *Material Journey* (*stages*), 2018: processes displayed) thus became not simply a 'finished' final artwork but also detailed the unseen energy that went into its production evidenced in not only the carbon footprint calculations but also the visual representation of four distinct stages of its production in the exhibition itself (Figure 3: *Material Journey* (*boat*), 2018: exhibition shot). The making of a glass cast object can be identified by its six distinct stages—master model

making, rubber mould making, wax modelling, plaster mould making and finally glass casting followed by the finishing stage of grinding and polishing. The carbon footprint calculations relied on available data, harvested from datasheets issued by manufacturers and from data sets produced by industry organisations. The carbon footprint calculation was divided into three distinct calculations. First, the direct emissions created by the materials used, including waste, and the energy used by equipment used in the making process. In the process of calculating the carbon footprint, I relied on the engineering expertise of freelance researcher Tom Jordan to assist. Although seven main gases are frequently produced in manufacturing processes, $CO_2$ is the largest and main gas emitted and that is what we concentrated on (see Supplementary 2: Carbon Footprint Calculation *Material Journey* Excel File). In the first scope, we considered the *direct* embodied energy for the immediate materials used in the production of the work itself (glass, molochite, plaster, wax, rubber silicone, water) which we estimated to be 55.08 kilos of $CO_2$. Secondly, we considered that the amount of $CO_2$ created from the waste of the project was negligible as the waste itself was to become part of the project itself. Thirdly, we considered the carbon footprint of the equipment used in the production of the piece. This included the use of the car to travel to and from the studio to the National Glass Centre, the use of the equipment such as the kiln, wax steamer, and rubber and wax melting pot. The carbon footprint of the car was by far the biggest of all $CO_2$ emissions of the actual equipment use itself. We estimated the carbon footprint of the equipment use to be 46.87 kilos of $CO_2$ of which the car was 46.76 kilos. In the second scope, we considered the *indirect* emissions of energy from its production—the firing of the kiln, the melting of the rubber compound and the wax, the lighting and heating of the studio and the gallery where the work was to be exhibited for two months. This, we estimated to be 323.74 kilos of $CO_2$. Finally, in the third scope, we calculated the indirect emissions from other sources, mainly emissions from transport—from the packaging in which the materials were delivered to the materials themselves, delivered by courier diesel vans. This, we estimated added another 2.9 kilos of $CO_2$. Overall, we estimated that this project generated 428.62 kilos of $CO_2$. An average tree can absorb 22 kilos of $CO_2$ per annum. Therefore, we argued that this project would need 19.5 trees to absorb its carbon footprint in a year.

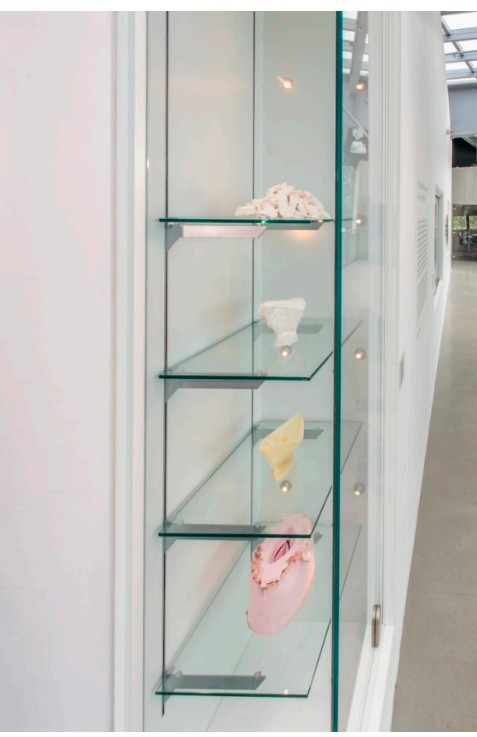

**Figure 2.** *Material Journey* (*stages*), 2018: processes displayed.

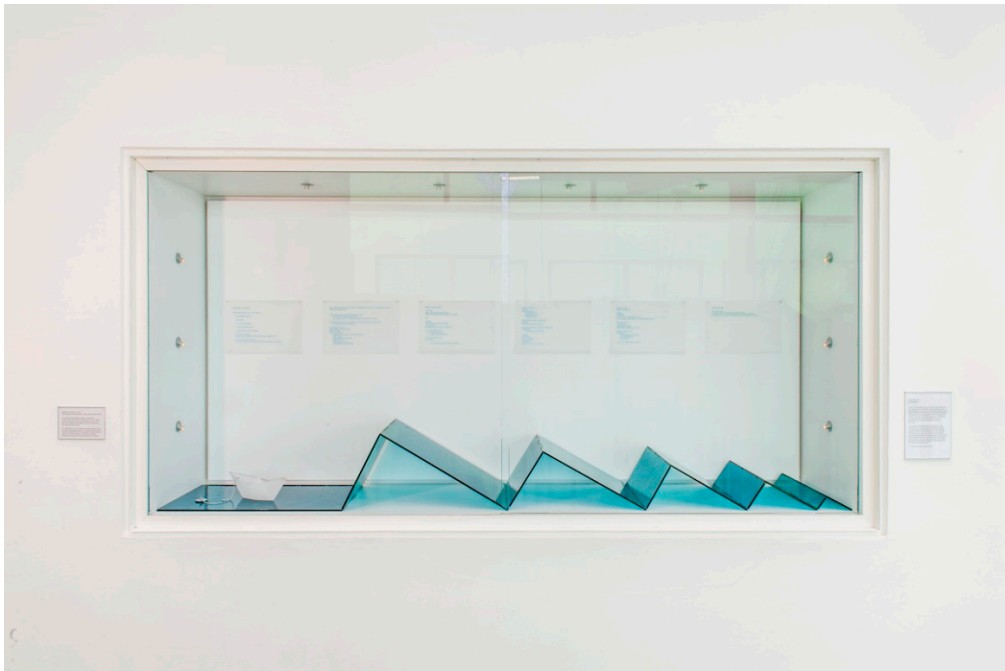

**Figure 3.** *Material Journey*, 2018—final work with a cast glass boat with a rubber plug hole atop a fused glass wave, with poem text in the background.

## 3. Conclusions

Advice from Creative Carbon Scotland, before we set out on this journey, included 'know the beginning before the end'. What became clear, is that there are no clear guidelines, or tools with which to calculate the carbon footprint of this simple art project. We relied on data provided to an extent by the industry. As expected, the actual firing of the work itself (259.99 kilos) accounted for a significant proportion of the carbon footprint. The actual carbon footprint of the materials themselves was second (55.08) closely followed by transport (46.78) which was the third largest $CO_2$ emitter. Agencies such as Creative Carbon Scotland and its English counterpart, Julie's Bicycle, provide some practical help and tools online for helping arts organisations reduce their carbon footprint. However, actual tools for calculating the material footprint of an art project remained an educated guesswork. It became clear also that this is, in part, a futile exercise—there are no clear boundaries as to where a material journey begins and where it ends. $CO_2$ emissions are currently calculated, and responsibility assigned to the country of origin, but as soon as things are shipped overseas, there is no national responsibility for those emissions accrued during transit. Its entangled messiness is perhaps emblematic of the Anthropocene. It could be argued that this project could be 'resolved' by firing a kiln run on renewable energy only, and driving an electric car to and from the studio and planting trees to offset its $CO_2$ production. However, it does not deal with the fundamental problem of the utter enmeshment of materials and processes of the human world with the natural world. What became clear is that the tools for understanding and calculating the carbon footprint of a material practice are currently lacking and need more development.

Secondly, the project concluded that even though the calculation of a carbon footprint is a useful and necessary exercise, if we are to understand the impact of our material practices better, it remains an isolated exercise if not part of a broader re-framing of the economic model itself. *Material Journey* will remain a material entity. Although purchased and now in the permanent collection of the National Glass Centre, the 'thing' that is the *Material Journey,* lives on in its material manifestations of its rubber mould (which can be re-used) melted wax (that can be recycled), waste mould materials (gone to the waste disposal) and waste glass (presumably gone to landfill but could be recycled). Much like ecosystems services thinking puts a value on a tree (for its oxygen producing capacity), which in

itself is futile if it is not accounted for in the current economic model (no one 'pays' for the tree), so does the carbon footprint calculation prove inadequate. However, if we are required to have a fundamental shift in thinking about what materials 'do', in order to fit into a circular, closed loop necessary for a doughnut economy, then both exercises provide a means of thinking differently about the materials and processes which underpin human existence in the Anthropocene. And that, of course, will include glass.

**Supplementary Materials:** The following are available online at http://www.mdpi.com/2076-0752/8/1/7/s1. Supplementary 1: *Material Journey* (2018). This text accompanied the cast glass work produced for Material Journey and is an integral part of the work. Supplementary 2: Carbon Footprint Calculation of Material Journey—Excel (2018).

**Funding:** This project received funding from the National Glass Centre to support the making of this new work.

**Acknowledgments:** The author wishes to thank the freelance researcher Tom Jordan for his contribution to the project with calculating the carbon footprint. All images reprinted courtesy of the artist and Kevin Greenfield (photographer). Creative Carbon Scotland was useful is giving direction to the research. Thanks also to the staff of the National Glass Centre who provided both useful data on the carbon footprint of the Centre and gave logistical help in setting up the exhibition.

**Conflicts of Interest:** The author declares no conflict of interest.

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
