# Peer review of "Glass—A Material Practice in the Anthropocene"

_arts, 2018_

Reviewer 1 Report

This is a timely and thoughtful examination of art/craft glass practice. It reads well with good references that construct a strong argument for further research.

Some small but important changes below.

Please change ‘carbon’ to CO2 at lines 133, 139 (…kilos of carbon), 143, 146, 147, 159, 164, 169 – the gas is referred to as CO2 (line 129) and proceeds to be described as ‘carbon’. Although often used as a shorthand substitute for CO2 it is a different material that is not considered to be a greenhouse gas - this undermines the writer’s authority. The term 'carbon footprint' is acceptable as it is not a description of a material, rather a concept or measurement.

Please change CO2 to CO2 (subscript 2) at line 129.

Author Response

Thank you for your thoughtful comments.

I have taken your observations on board about being careful with the terminology and will amend.

Thank you.

Reviewer 2 Report

The paper is of a very relevant and important topic. However, it is felt that the language is at times too complicated and entangled with long sentences and unnecessary words (e.g. line 74; temporary or ephemeral virtually means the same). Other examples are line 26; It is longwinded and complicated, but highly relevant and important for the paper. The following sentence (line 30) is far better.

If this paper is to create an impact with a wider audience as well as our next generation of makers and future students/researchers, it could benefit from an easier language. 

It needs a separate paragraph with conclusion and if possible, recommendation of future improvement of calculation of carbon footprint in the craft industry.

Author Response

The paper is of a very relevant and important topic. 

Thank you.

However, it is felt that the language is at times too complicated and entangled with long sentences and unnecessary words (e.g. line 74; temporary or ephemeral virtually means the same). 

I take your point that these two words temporary or ephemeral are very similar. However, to me they denote two different types of work; Andy Goldsworthy’s Rain Shadow works for example are very ephemeral; they evaporate in minutes. The Full Farm (1974) by the Harrisons for example is temporary but not ephemeral. I hope that clarifies that? 

Other examples are line 26; It is longwinded and complicated, but highly relevant and important for the paper. The following sentence (line 30) is far better. 

I have broken the long sentence up into four separate ones to make it easier to read; each sentence sums up a key finding.

If this paper is to create an impact with a wider audience as well as our next generation of makers and future students/researchers, it could benefit from an easier language. 

I am happy to re-write in for a different audience but was under the impression that the Arts Journal is predominantly for an academic audience?

It needs a separate paragraph with conclusion and if possible, recommendation of future improvement of calculation of carbon footprint in the craft industry.

I have done that. I have made the conclusion more concrete in that better tools are needed to enable practitioners to calculate their carbon footprint.

Thank you for your time and comments.

Much appreciated.